# A Comparison of Dinoflagellate Thiolation Domain Binding Proteins Using In Vitro and Molecular Methods

**DOI:** 10.3390/md20090581

**Published:** 2022-09-18

**Authors:** Ernest Williams, Tsvetan Bachvaroff, Allen Place

**Affiliations:** Institute for Marine and Environmental Technologies, University of Maryland Center for Environmental Science, 701 East Pratt St., Baltimore, MD 21202, USA

**Keywords:** dinoflagellate, algal toxins, biosynthesis, natural product, PKS, in vitro assay

## Abstract

Dinoflagellates play important roles in ecosystems as primary producers and consumers making natural products that can benefit or harm environmental and human health but are also potential therapeutics with unique chemistries. Annotations of dinoflagellate genes have been hampered by large genomes with many gene copies that reduce the reliability of transcriptomics, quantitative PCR, and targeted knockouts. This study aimed to functionally characterize dinoflagellate proteins by testing their interactions through in vitro assays. Specifically, nine *Amphidinium carterae* thiolation domains that scaffold natural product synthesis were substituted into an indigoidine synthesizing gene from the bacterium *Streptomyces lavendulae* and exposed to three *A. carterae* phosphopantetheinyl transferases that activate synthesis. Unsurprisingly, several of the dinoflagellate versions inhibited the ability to synthesize indigoidine despite being successfully phosphopantetheinated. However, all the transferases were able to phosphopantetheinate all the thiolation domains nearly equally, defying the canon that transferases participate in segregated processes via binding specificity. Moreover, two of the transferases were expressed during growth in alternating patterns while the final transferase was only observed as a breakdown product common to all three. The broad substrate recognition and compensatory expression shown here help explain why phosphopantetheinyl transferases are lost throughout dinoflagellate evolution without a loss in a biochemical process.

## 1. Introduction

Dinoflagellates are marine protists that can be readily split into two evolutionary groups, the basal syndiniales that are heterotrophic parasites and the distal “core” dinoflagellates [1,2,3] that are frequently mixotrophic with a complex evolutionary history with multiple chloroplast acquisitions and losses [4,5,6,7,8,9]. Most species, like other protists, produce many light sensing compounds, but they are also a source of many other natural products such as the polyunsaturated fatty acid docosahexaenoic acid (DHA) and compounds that can block ion channels associated with nerve function or create pores in membranes containing cholesterol [10,11,12]. Just as in bacteria and fungi, the majority of dinoflagellate natural products are made in a modular fashion whereby a carboxylic acid, usually acetic acid, but sometimes other small carboxylic acids or an amino acid, is attached to a carrier protein and chemically modified followed by another addition and modification and so on [13,14,15,16,17,18]. This is a deceptively simple means of biosynthesis, similar to protein synthesis, but with an almost limitless number of substrate and modification combinations giving rise to a huge diversity of compounds [10]. Even when limiting the search to dinoflagellates, many compounds have been discovered [19], usually noticed due to their impacts on human and animal health [12,20,21], leaving room for many more to be discovered. Indeed, maitotoxin from *Gambierdiscus toxicus* is the largest known natural product other than proteins and polysaccharides containing 98 stereocenters and 32 fused rings [22]. This makes industrial synthesis of these compounds unfeasible, and a need to understand the biosynthesis all the more pressing. 

Unlike in bacteria and fungi where the genes associated with natural product synthesis are grouped in the genome more or less in their order of operation [23,24,25], dinoflagellate genomes are frequently composed of many tandem repeats rather than cis-acting gene clusters [26]. The transcripts for natural product synthesis genes almost always contain a single domain with no other relevant genes nearby [27,28,29,30,31,32,33]. Moreover, the copy number of these genes in dinoflagellate genomes can be very high with over a hundred ketosynthases, the enzymes that bind and attach acetyl acid to a carrier protein, evident in many species [29,34]. Two genes with low copy number are thioesterases, enzymes that terminate synthesis by cleaving the compound from the carrier protein [13], and phosphopantetheinyl transferases (PPTases) that attach the phosphopantetheinate arm of coenzyme A to the carrier protein yielding a free thiol group upon which the growing compound can be added and removed as well as giving the carrier protein its name of “thiolation domain” [35]. The thioesterase gene was not chosen for this study because it acts to terminate a series of processes and given the size of the *A. carterae* toxin a traceback of intermediate products was not feasible, although knockouts in other systems have proven fruitful [36,37]. The PPTases, on the other hand are the first rate limiting step in biosynthesis, and are also the gatekeepers separating lipid synthesis—the most basic and iterative example of natural product synthesis—from all other pathways [38]. Frequently one or a few redundant PPTases are used for each compound made, and of lipid and other natural product synthesis, including polyunsaturated fatty acids, which are strictly separated [39,40,41]. Although the PPTases that make natural products can frequently rescue the function of each other, the PPTase for lipid synthesis is specific to the acyl carrier protein (ACP), the thiolation domain for lipid synthesis, and vice versa as exemplified by enterobactin and surfactin synthesis in *E. coli* and *Bacillus subtilis*, respectively [42]. Thus, assessing the specificity of dinoflagellate PPTases to various thiolation domain sequences can help identify natural product pathways and separate them from lipid synthesis.

This study builds on previous studies that have established the presence of three clades of PPTases and expressed the *Amphidinium carterae* PPTases heterologously in *E. coli* [43] as well as substituted thiolation domains into the BpsA gene [44] from *Streptomyces lavendulae* [45], an indigoidine synthesizing reporter previously used in in vitro reporter assays [46,47,48]. Specifically, PPTase and BpsA constructs expressed in *E. coli* or synthesized in vitro were purified for use in in vitro assays that use the amount of indigoidine production to determine the phosphopantetheination efficiency of each thiolation domain by each PPTase with the assumption that more indigoidine production is the result of more efficient binding, and that these PPTase/thiolation domain combinations are more likely to occur in-situ. In vitro methods have the advantage over previous methods looking at indigoidine production in *E. coli* in that the kinetics of the reaction as well as the molar amounts of the enzymes and substrates, which can, more easily, be controlled as well as the fact that the *A. carterae* acyl carrier protein cannot be expressed in *E. coli*, likely due to host toxicity [44]. The results of indigoidine production assays were double checked using a fluorescence based free thiol detection method to assess the amount of phosphopantetheinate added to each thiolation domain. Additionally, the expression patterns of each PPTase during a growth curve of the dinoflagellate *Amphidinium carterae* were measured by western blot and sequence motifs were identified to try to assign a biological function to each PPTase when compared to the results of the in vitro experiments. The broad substrate specificity, or lack thereof, combined with the oscillating expression patterns of *A. carterae* PPTases shown in these experiments is certainly different from what is canonical in bacteria and fungi, but it also explains many of the other unique features of natural product synthesis in dinoflagellates such as frequent gene gains and losses as well as gene copies without evident protein expression.

## 2. Results

### 2.1. PPTase Activation of Thiolation Domains

All three PPTases [43] were able to activate indigoidine production in the wild-type BpsA reporter in a dose dependent manner (Figure 1). The same was true for thiolation domain four of the triple KS transcript (3KS4) with a similar change in rate versus changes in reporter concentration but with an approximately ten-fold decrease in absorbance at 590 nm. In both cases the Clade three PPTase correlated with a slightly higher rate of indigoidine production followed by Clade two and then Clade one with low error between replicates. Reporters containing thiolation domain one of both the BurA-like (BurA1) and the ZmaK-like (ZmaK1) transcripts did not appear to make appreciable amounts of indigoidine during the time of these experiments for any of the three PPTases.

Phosphopantetheination of the BpsA reporter by all three PPTases was also evident by the fluorescent free thiol detection assay (Figure 2). Although the Clade one and two PPTases had the highest levels of free thiol (phosphopantetheinate) detected at the same pH as the indigoidine synthesizing assays, around pH 7.5, the Clade three PPTase had peak fluorescence at pH 6.5. Not surprisingly, the observed phosphopantetheination of the reporter containing the 3KS4 thiolation domain was much larger with the Clade three PPTase than either the Clade one or two PPTases when performed at the optimum pH (Figure 3). This was quantified using a CoA standard curve (Figure 4) giving a final concentration of CoA attached to the BpsA protein in µM. Relative to the 8 µM starting material of BpsA gives a phosphopantetheination level for the 3KS4 domain of approximately 50% for the Clade three PPTase and approximately 12.5% for the Clade one and two PPTases. Despite a lack of evident phosphopantetheination in the indigoidine production assays for the BurA1 and ZmaK1 thiolation domains, phosphopantetheination was evident in the free thiol detection assay for these domains for all three PPTases but with a larger error. Moreover, the Clade three PPTase gave the lowest yield with approximately 10% of starting material phosphopantetheinated while the Clade one and two PPTases yielded approximately 18% phosphopantetheination. Phosphopantetheination was also evident using the free thiol detection method for the acyl carrier protein (ACP), the thiolation domain responsible for lipid synthesis (Figure 5). Again, all three PPTases were able to phosphopantetheinate the thiolation domain with the Clade two PPTase yielding the highest level with approximately 75% of the 15 µM starting material phosphopantetheinated followed by Clade three at 30% and Clade one at 23%. This free thiol detection method cannot be compared to the indigoidine synthesis assay for the ACP since the ACP containing BpsA gene was not expressible in *E. coli*.

### 2.2. PPTase Biology

Cell counts for the growth curve demonstrate sampling at the transition from lag to log phase at day 0 through log phase until day 16 and into stationary phase (Figure 6).

The cell counts ranged from approximately 30,000 cells/mL to 145,000 cells/mL with pH maintained at 7.8 ± 0.2. Total protein per 400,000 cell aliquot rose throughout the log phase, but then dropped upon entry into stationary phase. The acyl carrier protein expression seemed to have an opposite trend with a reduction upon entry into log phase, followed by an increase prior to entry into stationary phase, and then a plateau. The Clade one and three PPTases seemed to have opposing expression Clade three expression giving way to stable Clade one expression, followed by an absence of Clade one and high expression of Clade three on day 14, and, finally, a short period of Clade one expression followed by Clade three expression. There were six time points where both the Clade one and three PPTases were expressed: days 0, 2, 9, 14, 19, and 21; and on day 16 neither were expressed. The Clade two PPTase was never expressed in its whole form giving rise to the observation of a consistent hypothetical breakdown product (Figure 7). Similar sized bands are also visible for the Clade one and three PPTases at approximately the same times as the full-sized bands. Given the location of the epitope the breakdown product would correspond to cleavage between the final two alpha helices that make up the core of each enzyme (Figure 8).

### 2.3. Phosphopantetheinyl Transferase and Acyl Carrier Protein Sequence Analysis

The 5′ and 3′ ends of all three *Amphidinium carterae* PPTases were successfully sequenced with evidence of the spliced leader. The Clade two PPTase does not appear to have a stop codon, while Clades one and three have a canonical dinoflagellate open reading frame (Genbank accession numbers ON157050-ON157050). The 5′ ends of all three PPTases are similar in length (51, 50, and 56 bases for Clade one, two, and three, respectively), but the 3′ ends are different between Clade one and two compared to three with Clade three having the longest 3′ untranslated region (UTR). The ten bases before and after the Clade one PPTase stop codon are 86% similar to the same position in the Clade two sequence, and if a stop codon is assumed in the Clade two PPTase the resultant 3′ UTR is very similar in folding structure to the Clade one UTR compared to the Clade three UTR, although with greater free energy (Figure 9).

SignalP as well as WolF PSORT did not detect localization signals for the three PPTases except for a weak cytoplasmic signal for the Clade three sequence whereas the acyl carrier protein sequence used here (ACP) has a strong chloroplast target sequence. Ubiquitination sites were evident for all three PPTases with the Clade three PPTase yielding the highest score and with multiple sites detected on every PPTase. The Clade three PPTase was also the only protein predicted to be stable.

## 3. Discussion

This work is the first example of heterologously expressed dinoflagellate proteins used in an in vitro assay. It is also the first example of a catalytically active dinoflagellate protein produced by an in vitro synthesis method. This is quite exciting given the general difficulties in heterologous protein expression [49,50], but also means that codon optimization is no longer required [51] allowing for the use of native transcripts. *E. coli* was used successfully in this study for protein expression, but mammalian cells have also been used [52] indicating that eukaryotic cells can also be utilized for proteins that require cleavage, chemical modifications, or to test cellular localization predictions. The use of in vitro and in vivo methods will hopefully increase our understanding of dinoflagellate biology where common techniques such as promoter modification and targeted genetics have been hampered by a largely post-transcriptional control of gene regulation and very high gene copy number [26,53,54,55].

Gene knockdowns and knockouts are the most common method for modifying protein expression in dinoflagellates [56,57], and are likely the methods, moving forward, for understanding natural product synthesis in dinoflagellates, despite the difficulties presented by high copy number. This study exploits the stable chemical modification of one actor in natural product synthesis by another. This allowed us to test the interactions of two proteins indirectly by measuring indigoidine synthesis or directly by measuring the increase in free thiol groups following the attachment of the phosphopantetheinate group to the thiolation domain. In the downstream reactions the biosynthetic enzymes interact with the chemical being synthesized either directly or at the site of attachment to the phosphopantetheinate group making specific interactions much more difficult to discern. With dozens of ketosynthases to choose from, trying to figure out which one attaches which exact acetate in a molecule like amphidinol with sixty-five carbons [58] is certainly daunting. Thus, indirect methods may be more useful such as targeting acyl transferase and thioesterase domains that, as noted by Van Wagoner et al. [19], are more likely to have a recognizable impact on the final structure of dinoflagellate toxins and potential inhibitors exist [59,60,61]. They are also generally low in copy number similar to the PPTases used in this study with an average of twelve thioesterase domains in dinoflagellate transcriptomes [34]. In addition to being the target of PPTases, the thiolation domains likely involved in natural product synthesis have been shown to have six to seven tetratricopeptide repeats [62],unlike the acyl carrier protein (ACP) for lipid synthesis, that may serve to scaffold biosynthetic complexes. This could allow for the enrichment of catalytic complexes for natural product synthesis from protein extracts using antibody-based methods. Essentially, a bit of whittling down is necessary before further biochemical validation of the roles of dinoflagellate natural product genes is feasible.

The differences between the indigoidine production and the free thiol detection assays show a disconnect between the ability of the BurA and ZmaK insert reporters to produce indigoidine and their ability to be phosphopantetheinated (Figure 3 and Figure 5). The lack of indigoidine production is likely due to steric inhibition from the thiolation domain that must be positioned to interact with all other domains for successful biosynthesis. This is similar to previous results where the phosphopantetheination of this reporter was performed in *E. coli* [43] and may help explain why there is a reduction in indigoidine synthesis with the 3KS4 thiolation domain versus the original BpsA domain (Figure 1), but with a similar change in rate with increasing substrate concentration. This makes the indigoidine production assay useful for providing yes/no type answers as has been suggested before [48], but requires confirmation of negative results. The free thiol assay on the other hand is a direct measurement and is likely at least semi-quantitative. In order to be adapted to other types of natural product synthesis interactions a detectable chemical change is required, which is unfortunately hard to come by. Modified analogs or radioisotopes have also been used in the past for PPTase [63,64,65], but may be difficult to extend to the downstream biochemistry of natural product synthesis given that acetate, the dominant substrate incorporated into dinoflagellate toxins, is used in so many other biological processes. Development of an in vitro assay would alleviate this but may prove tedious given the large number of ketosynthases in dinoflagellate genomes, although there is evidence of expression changes during nitrogen starvation indicating that they may be key in regulating natural product synthesis and they are more diverse giving more power to these sorts of analyses [66].

The results of the free thiol assay show that each of the three PPTases can phosphopantetheinate all of the thiolation domains used in this study, including the acyl carrier protein at near equivalent amounts (Figure 5 and Figure 6). This is quite interesting given the ambiguity of the origins of these genes with thiolation domains only present in *Hematodinium* and PPTases absent from every sequenced syndinian dinoflagellate. Taken together with the biological data from sequence analysis and protein abundance during a growth curve gives two very unusual interpretations. First is that despite being biologically active when a stop codon is introduced, the Clade two PPTase is retained in the genome without having an apparent stop codon and without evidence of expression as an intact protein. This is especially surprising considering that the Clade two PPTase was able to phosphopantetheinate the ACP, required for the vital process of lipid synthesis, to a greater extent than the other two PPTases (Figure 5). The Clade one and two PPTases are very similar in sequence, including the 3′ UTR (Figure 9), indicating that they may be the result of gene duplication with a loss of function for the Clade two sequence. The second surprising interpretation based on the apparent lack of PPTase specificity is that lipid and natural product synthesis are not segregated based on PPTase targets, as is usually the case in bacteria and fungi [35,40]. The most basally branching dinoflagellates are the syndinian clades [1] that are not photosynthetic but instead parasitize other eukaryotes. Although not well studied, there is some genetic information including genomic data for *Ameobophyra ex. Karlodinium veneficum*, a parasite of the dinoflagellate *Karlodinium veneficum* that does not have any lipid synthesis machinery and presumably gets its lipids from the host [67]. This includes the PPTase that is generally assumed to be present in all life since lipid synthesis is usually required for biological life to exist. Another species of syndinian dinoflagellate is *Hematodinium sp.*, a crustacean parasite that also does not have an identifiable PPTase but does possess the triple KS transcript from which one of the thiolation domains in this study is derived [68]. Thus, it is unlikely that dinoflagellates have a native PPTase and the PPTases that are evident may have been acquired through horizontal gene transfer, possibly from the chloroplast as has happened in the other cases [5,6,9]. This may explain the lack of binding site specificity observed with these PPTases as well as other protists [64]. Whether specificity was present when acquired and deteriorated over time or if the PPTase acquired during endosymbiosis already lacked specificity is unclear. The “three Clade” definition itself is largely arbitrary since there is little phylogenetic context to put these genes in given the lack of sequence data available in protists [43]. The benefit of the lack of substrate specificity is that natural product genes can be acquired over time and are more likely to be utilized if there is an existing PPTase that can activate the biosynthetic pathway. This may help to explain the likely horizontal gene transfer of the BurA-like and ZmaK-like genes that are not evident in any of the basal syndinian lineages but are common in most core dinoflagellate lineages. A benefit to lipid synthesis may also be reaped from redundant copies by ensuring that any expressed ACP, the thiolation domain for lipid synthesis is always present in the holo form. Moreover, while there are three PPTases in *Amphidinium carterae*, a basal core dinoflagellate, there are many derived species that have lost PPTases, including *Protoceratium reticulatum*, which only has the Clade three PPTase [43]. Thus, natural product synthesis in dinoflagellates may be so dynamic compared to bacteria and fungi because the biosynthetic capabilities have been acquired by horizontal gene transfer.

## 4. Materials and Methods

### 4.1. Protein Expression and Purification

The *E. coli* BL21(DE3) cells containing pET-20b plasmids with each of the three codon optimized PPTase sequences from the basal toxic dinoflagellate *Amphidinium carterae* (Hulbert) [43] were grown in autoinduction media [69] with 100 µg/mL carbenicillin at 25 °C for three days at 250 rpm. Likewise the pCDFDuet-1 plasmid containing the BpsA gene [48] was modified to contain four thiolation domains from four *A. carterae* transcripts: (1) the triple KS (thiolation domain 4), (2) BurA-like (thiolation domain 1), (3) ZmaK-like (thiolation domain 1) genes [27,33,70,71], as well as the (4) acyl carrier protein (ACP), and transformed into chemically competent BL21(DE3) cells (Thermo Fisher, Waltham, MA, USA) according to the manufacturer’s instructions. These particular thiolation domains were chosen to represent the variety of sequences at the phosphopantetheination site (Figure 10). These constructs could not be expressed by autoinduction since production at temperatures above 20 °C did not produce active BpsA protein capable of producing indigoidine. Instead, 10 mL LB media containing 50 µg/mL spectinomycin was inoculated with each of the dinoflagellate thiolation domain containing *E. coli* clones and grown overnight at 37 °C at 250 rpm in a 250 mL Erlenmeyer flask. These cultures were then diluted to 500 mL LB with antibiotics and grown to an OD 600 of 6.0. Cultures were then chilled to 18 °C in an ice bath, induced with 500 µL 0.1 M IPTG, and grown overnight at 18 °C at 250 RPM. 

Induced *E. coli* clones were collected after protein expression by centrifugation at 10,000× *g* for 15 min at 4 °C, and the supernatant was decanted. Pellets were stored at −80 °C until processing. Frozen pellets were suspended in 25 mL of lysis buffer containing 5 mM imidazole, 500 mM NaCl, 25% (*v/v*) glycerol and 20 mM Tris/HCl pH 7.5 along with bacterial protease inhibitors (Sigma Aldrich, St. Louis, MO, USA) and thawed at 4 °C. Cells were lysed with a French press chilled to 4 °C with 1000 PSI at the piston and 20,000 PSI at the outlet. The lysate was clarified by centrifugation at 12,000× *g* for 15 min at 4 °C and the supernatant was treated with benzonase (Thermo Fisher, Waltham, MA, USA) overnight at 4 °C. The His-tagged lysate was bound to a 1 mL Hi-trap crude cobalt column (Cytiva, Marlborough, MA, USA); washed with 25 column volumes of 5 mM Imidazole, 250 mM NaCl, 20 mM Tris/HCl pH 7.5; and eluted into 10 separate volumes of 250 mM Imidazole, 250 mM NaCl, 20 mM Tris/HCl pH 7.5, and 12.5% glycerol using an AKTA chromatography system (Cytiva). Elution fractions were separated with 4–12% Bis-Tris gels from Novex (Thermo Fisher) in MOPS buffer at a constant voltage of 165 V for 50 min and stained with Imperial stain (Thermo Fisher) according to the manufacturer’s directions to verify protein capture. Fractions containing expressed protein were concentrated using a 10 mL Amicon high pressure stirred filter apparatus with a 5 kD regenerated cellulose filter for the PPTases and 30 kD filters for the BpsA constructs at 45 PSI with Nitrogen at 4 °C (Sigma Aldrich). Proteins were then buffer exchanged with 50 mM TRIS pH 7.5 using an amicon 500 µL spin column with a 3 kD nominal pore size for the PPTases and 10 kD for the BpsA constructs at 4 °C (Sigma Aldrich). The BpsA vector containing the acyl carrier protein (ACP) thiolation domain could not be expressed in *E. coli* as discussed in [44] and the PPTase from dinoflagellate Clade two was retained in the insoluble pellet following lysis. For these proteins, the Pure Express kit (New England Biolabs, Ipswich, MA, USA) was used with approximately 10 ng of pET-18b plasmid containing either the PPTase Clade two or the entire ACP open reading frames according to the manufacturer’s direction for synthesis in vitro. This means that the ACP thiolation domain was not expressed in the BpsA framework and was not used for indigoidine based detection of phosphopantetheination by the three dinoflagellate PPTases, but was available for the direct measurement of free thiol. Artificial ribosomes from the Pure Express kit were removed by passage through a 100 kD Amicon spin column at 10,000× *g* for 20 min followed by buffer exchange and concentrated using a 3 kD Amicon spin column with two washes at 500 µL each in the same manner as the purified PPTases expressed in *E. coli*. 

Purified protein in 50 mM TRIS was prepared for storage by the addition of glycerol to a final concentration of 12.5% and quantified using a Q-bit protein quantification kit (Thermo Fisher). This was adjusted based on the band densities following imperial staining to account for spurious bands from His-tag purification.

### 4.2. In Vitro Indigoidine Production

Each of the three BpsA constructs containing the dinoflagellate thiolation domains along with the original BpsA thiolation domain were combined with each of the three dinoflagellate PPTases according to the protocols from Owen et al. 2011. Briefly, a premix containing one of each PPTase at 0.2 µM and one of each BpsA construct in two-fold dilutions from 8 µM to 0.25 µM in 50 mM Tris pH 7.5, 10 mM MgCl2, and 100 µM CoA was incubated at 30 °C for 10 min. This was then combined with a ⅓ volume of 25 mM glutamine and 5 mM ATP for a final volume of 50 µL. The absorbance at 590 nm was measured every 10 s for 30 min to quantify the indigoidine production using a Spectramax i3x plate reader (Molecular Devices, San Jose, CA, USA). Five minute windows of linear indigoidine production were used to determine the maximum rate of production for each combination of PPTase and BpsA constructs.

### 4.3. Free Thiol Quantification

The addition of phosphopantetheinate was quantified using the free thiol detection kit from Abcam (Cambridge, UK). A seven-point standard curve was performed using 5-fold dilutions of CoA in 50 mM TRIS pH 7.5 according to the directions of the kit with 50 mM TRIS pH 7.5 as a blank control (Figure 4). Linearity of detection was evident 50 µM to 170 nM CoA. The pH optimum of each of the three PPTases was determined by buffering triplicate reactions in 0.5 unit increments from pH 5.0 to 5.5 using MES, 6.0 to 6.5 using HEPES, and from 7.0 to 8.5 using TRIS at 50 mM each. Reactions were set up as follows: 50 µL total volume with 50 mM buffer, 100 µM CoA, 10 mM MgCl2, 0.2 µM of one of the three PPTases, 4 µM of the BpsA wild-type reporter, and 5 mM ATP with a 10 min pre-incubation at 30 °C prior to ATP addition. The reaction was allowed to proceed for 20 min at 30 °C and halted with 250 µL of ice cold 2 M NaCl, 50 mM Tris pH 7.5. Free CoA was removed by passage of the halted reaction through a 3 kD Amicon spin filter at 10,000× *g* for 15 min at 4 °C followed by two washes with 250 µL of 50 mM Tris pH 7.5. The amount of phosphopantetheinate was then determined using the free thiol detection kit according to the manufacturer’s directions for each of the triplicate reactions along with three 10-fold dilutions of CoA starting at 25 µM to compare to the standard curve as well as a blank reaction. A pH of 7.5 was chosen for PPTases from Clade one and two and 6.5 was chosen for the Clade three PPTase based on these empirical results for all subsequent reactions. (Figure 2). 

Phosphopantetheination reactions were repeated at optimum pH for each of the three PPTases at 0.2 µM along with the BpsA reporter containing thiolation domains 4 from the triple-KS transcript (3KS4), 1 from the BurA-like transcript (BurA1), and 1 from the ZmaK-like transcript (ZmaK1) at 4 µM, as well as the acyl carrier protein (ACP) at 10 µM in triplicate along with negative controls without CoA added. These reactions were purified using a 3 kD Amicon filter and free thiol was measured using the free thiol detection kit along with three CoA standards and blank controls in the same manner as the pH optimization protocols. Assuming that only one phosphopantetheinate group can be added to each thiolation domain, the amount of free thiol was used to determine the percent of thiolation domains phosphopantetheinated in 20 min.

### 4.4. PPTase Expression

An axenic culture of *Amphidinium carterae* (Hulbert) NCMA strain 1314 was prepared in 50 mL 32 ppt ESAW medium [72] modified to have 1 mM HEPES using 100 µg/mL carbenicillin, 50 µg/mL kanamycin, and 50 µg/mL spectinomycin and grown at 20 °C on a 14/10 day night cycle at 50 microeinsteins illumination. Upon reaching approximately 30,000 cells/mL the culture was diluted 1:2 every Monday, Wednesday, and Friday until a volume of 1 L was reached. The culture was then transferred into a 20 L multiport polycarbonate vessel with aeration supplied through a 0.2 µm filter. Dilution continued every Monday, Wednesday, and Friday using sterile media without antibiotics to a volume of 18 L and a density of 30,000 cells/mL. pH was maintained at 7.8 ± 0.2 using a pH controller with a solenoid attached to a CO2 cylinder bubbling at a rate sufficient to correct the pH by 0.2 units in approximately 1 min. Then, 500–1000 mL of high density and low density culture, respectively, was harvested by centrifugation at 1000× *g* for 10 min at 4 °C on days 0, 2 5, 7, 9, 12, 14, 16, 19, 21, 23, and 26. Cell counts were taken and the cell pellet was diluted in 2× SDS-PAGE sample buffer to a concentration of 40,000 cells/µL.

A total of 100 ng purified protein for quantification purposes and 10 µL (400,000 cells) of total protein from each time point along with a Seeblue plus2 pre-stained ladder were separated on a 4–12% Bis-Tris gel (Novex, Waltham, MA, USA) in MOPS buffer at 165 V for 50 min for western blotting with antibodies to each of the three PPTases and the ACP (Table 1, Figure 11). Separated proteins were transferred to PVDF membranes using the Trans-Blot Turbo Transfer system from BioRad (Hercules, CA, USA) according to the manufacturer’s settings for a standard 1 mm gel. Total protein transferred was quantified using the AzureRed total protein stain from VWR (Radnor Township, PA, USA) and imaged on an Azure imaging system according to the manufacturer’s protocols. Blots were blocked and exposed to primary and secondary antibodies using the iBind system (Thermo Fisher) with a 1:500 dilution of each primary antibody (Genscript, Piscataway NJ, USA) and a 1:50,000 dilution of a horseradish peroxidase conjugated goat anti-rabbit secondary antibody (BioRad). Westerns were imaged using the SuperSignal West Pico chemiluminescence kit from Thermo Fisher on a BioRad ChemiDoc system with optimal exposure and 4×4 binning. The pre-stained ladder was also imaged, and the two images were merged in BioRad’s Image Lab software version 6.1.0. Bands were identified based on relative molecular weight and quantified based on band density. The conversion of band density to nanogram estimates was based on a comparison of the 100 ng standard from each blot to a standard curve using 3-fold dilutions of purified protein and a power equation describing the relationship between concentration and band density from https://www.dcode.fr/function-equation-finder (accessed on 3 March 2022) that was modified to better fit data at lower concentrations. This was not performed for the Clade two PPTase since the full-size protein was never observed. 

### 4.5. PPTase Sequence Analysis

Transcripts of the three *A. carterae* PPTase originally described in [43] were used to design primers for full length sequencing. Primers were designed to amplify the 5′ and 3′ untranslated regions (UTRs) based on the open reading frames of each PPTase along with the spliced leader [73], a low variability sequence spliced onto each messenger RNA in dinoflagellates, and a poly-T primer with a GC lock and a priming sequence not found in the *A. carterae* transcriptome (Table 2). RNA was isolated from *A. carterae* cultures using Tri-reagent (Sigma Aldrich, St. Louis, MO, USA) according to the manufacturer’s directions and reverse transcribed using Superscript II reverse transcriptase (Thermo Fisher) with 50 nM poly-T primer and 5 nM of each reverse primer from the PPTase open reading frames according to the directions of the reverse transcriptase. Amplification of cDNA template was performed using the Phusion high fidelity polymerase (New England Biolabs, Ipswich, MA, USA) with final concentrations of 1 ng/µL of template and 500 nM of forward and reverse primers for each PPTase for each reaction. Thermal cycling conditions consisted of an initial denaturation of 98 °C for 2 min followed by 35 cycles of denaturation at 95 °C for 15 s, annealing at 60 °C for 5′ UTR reactions and 68 °C for 3′ UTR reactions for 20 s, and extension at 72 °C for 1 min and a final polishing step at 72 °C for 5 min. This resulted in several bands when visualized on a 1% agarose gel in 0.5× TBE separated at 15 V/cm. Gel excision was performed using the Monarch Gel Extraction kit from New England Biolabs and the bands were sequenced on an Applied Biosystems 3130XL fragment analyzer at the Bioanalytical Services Lab (BasLab) in Baltimore, MD USA. The 5′ UTR for Clade two and three PPTases were sequenced confirming the spliced leader as well as the 3′ UTR of the Clade three PPTase. The 5′ sequence of the Clade one and two PPTases were very similar and indicated that the Clade one PPTase sequence started just after the spliced leader negating a need for further sequencing. The 3′ ends of the Clade one and two PPTases were very difficult to sequence, and a BLAST analysis of the primers used against the *A. carterae* transcriptome showed many sequences with very high or identical similarity, despite a length of twenty bases. A thirty base length primer set was designed (Table 2) and the amplification and sequencing methods were attempted again resulting in 3′ sequence for both the Clade one and two PPTases.

The predicted folding structure of each of the PPTase 3′ UTRs was performed at the Fold Web Server (https://rna.urmc.rochester.edu/RNAstructureWeb/Servers/Fold/Fold.html, accessed on 15 December 2021). Subcellular localization motifs were predicted using WolF PSORT (https://wolfpsort.hgc.jp/, accessed on 9 February 2021) with the animal sequence database as well as SignalP (https://services.healthtech.dtu.dk/service.php?SignalP-4.1, accessed on 9 February 2021). Ubiquitination site prediction was performed with UbiSite (http://csb.cse.yzu.edu.tw/UbiSite/, accessed on 25 January 2021) using the high threshold cutoff.

## 5. Conclusions

The demonstration of using purified protein in in vitro assays is an important advancement in the search to understand dinoflagellate biology. They are such a strange and diverse group of organisms, in that comparisons to other species, especially many common models, can be misleading. Even in this study, the interpretation of the in vitro assays alone might lead one to assume that the Clade two PPTase is most likely responsible for activating lipid synthesis by phosphopantetheinating the acyl carrier protein. Looking at the protein expression gives the very unexpected result that this protein is seemingly immediately broken down making a biological role for this enzyme entirely unlikely apart from a possible regulatory agent. Indeed, this study utilized enzymes from *Amphidinium carterae*, but proteins from other species may give very different results and may have unique functionalities. Taking as holistic an approach as possible is important when dealing with such a dynamic group of organisms, and biochemical validation will certainly be an important tool in the future. This study also reveals the importance of taking the evolutionary history of dinoflagellates into account when interpreting data. Although the bulk of study is on the core dinoflagellates that are dominated by photosynthetic species, the common ancestor is likely heterotrophic and parasitic. This means that any plastid associated process is likely to have a very complicated evolutionary history and keeping an open mind is essential.

## Figures and Tables

**Figure 1 marinedrugs-20-00581-f001:**
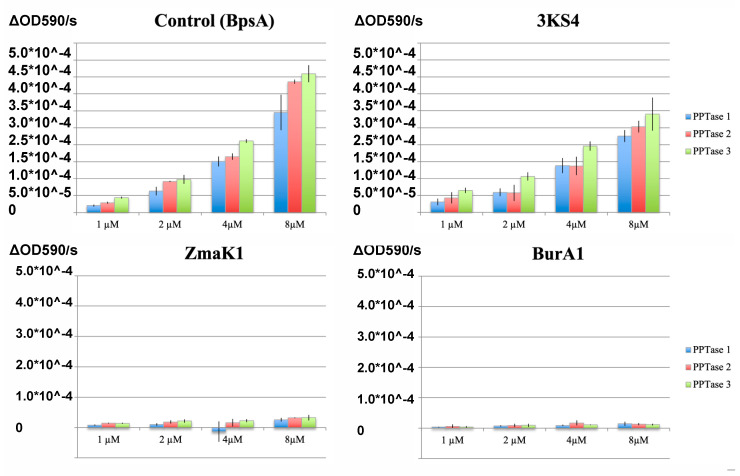
Kinetics of indigoidine production for combinations of thiolation domains and phosphopantetheinyl transfers (PPTases). The kinetics of indigoidine production are shown with the change in absorbance at 590 nm on the Y-axis and the concentration of each reporter on the X-axis of each graph titled with the thiolation domain inserted into the reporter. Error bars show the standard deviation for each set of reactions with PPTase clade one in blue, clade two in red, and clade three in green.

**Figure 2 marinedrugs-20-00581-f002:**
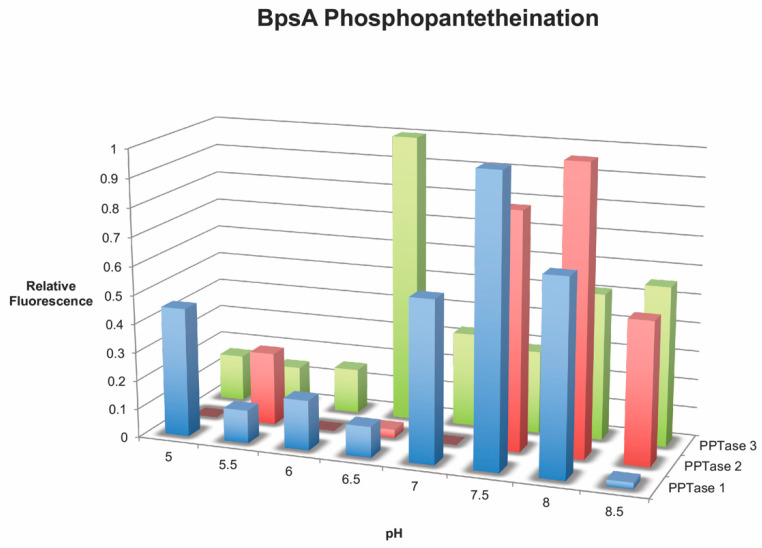
Phosphopantetheination of the BpsA reporter at various pH values. The graph shows the amount of phosphopantetheination of the wild-type BpsA reporter by each of the three *Amphidinium carterae* phosphopantetheinyl transferases. The relative fluorescence produced by the free thiol detection kit on the Y-axis as a function of pH in the reaction on the X-axis is indicative of the amount of phosphopantetheinate added to the BpsA protein by each phosphopantetheinyl transferase labeled on the Z-axis.

**Figure 3 marinedrugs-20-00581-f003:**
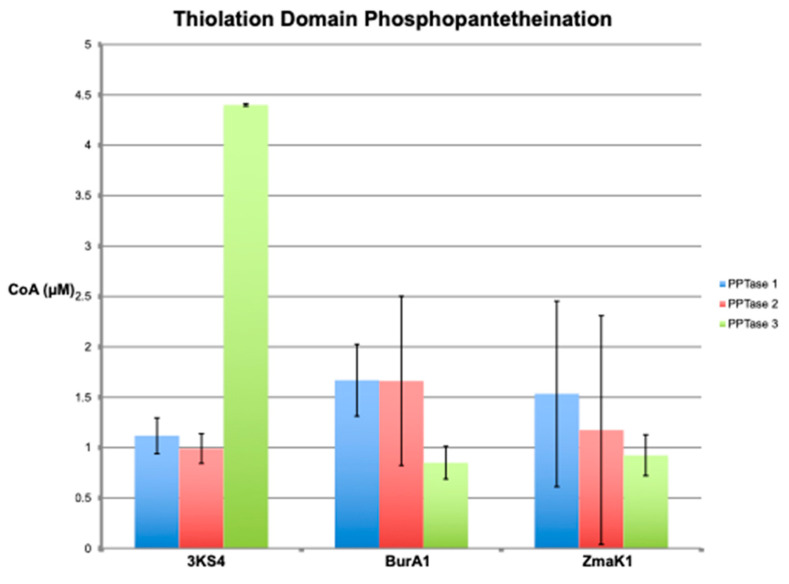
Phosphopantetheination of thiolation domains detected as free thiol. The graph shows the amount of phosphopantetheination of BpsA containing thiolation domain four from the triple KS transcript (3KS4), thiolation domain one from the BurA-like transcript (BurA1), and thiolation domain one from the ZmaK-like transcript (ZmaK1) described on the X-axis. The amount of phosphopantetheination is given as the resultant micromolar amount of CoA added to the 8 µM of starting BpsA protein. Error bars show the standard deviation of triplicate reactions for the Clade one, two and three PPTases colored blue, red and green, respectively.

**Figure 4 marinedrugs-20-00581-f004:**
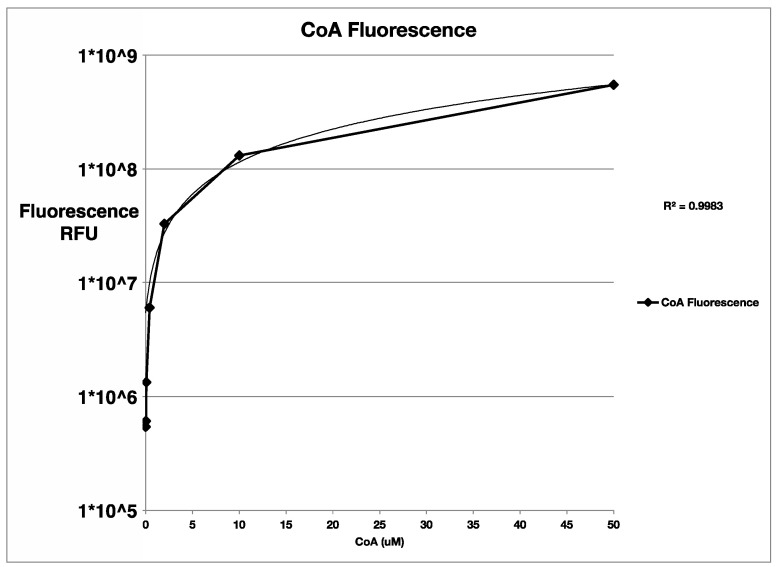
Standard curve of Coenzyme A (CoA) detected using a free thiol fluorescent assay. A standard curve is shown using a 5-fold dilution of Coenzyme A from 50 µM to 170 nm. The Y-axis shows the relative fluorescent units from the fluorescent free thiol detection kit following blank subtraction as well as a negative control, and the X-axis shows the Coenzyme A concentration in micromolar. A linear fit is also shown on top of the observed data as well as sum of the squared residuals on the right-hand side.

**Figure 5 marinedrugs-20-00581-f005:**
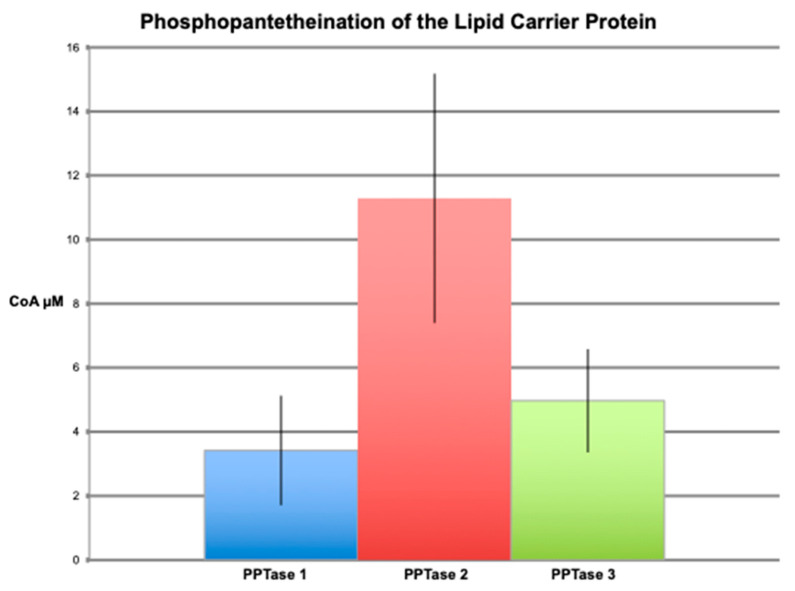
Phosphopantetheination of acyl carrier protein detected as free thiol. The graph shows the amount of phosphopantetheinatation of the acyl acrrier protein (ACP), the thiolation domain for lipid synthesis. The amount of phosphopantetheination is given as the resultant micromolar amount of CoA added to the 15 µM of starting ACP protein. Error bars show the standard deviation of triplicate reactions for the Clade one, two and three PPTases colored blue, red and green, respectively.

**Figure 6 marinedrugs-20-00581-f006:**
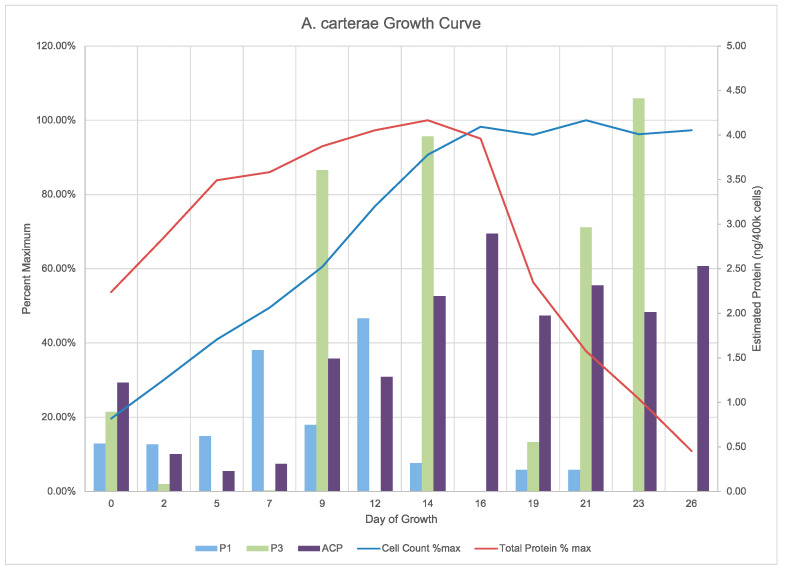
*Amphidinium carterae* growth curve cell counts and protein quantities. The graph shows the growth and protein measurements for an axenic *Amphidinium carterae* culture with CO2 addition. The cell counts (blue line) and total protein (red line) are shown as a percent of maximum on the left Y-axis while the western blot quantifications for the acyl carrier protein (ACP, purple) as well as the clade one and three PPTases (P1 blue, P3 green, respectively) are shown on the right Y-axis as an estimate of protein in ng/400k cells. The day of growth for each sample is shown on the X-axis.

**Figure 7 marinedrugs-20-00581-f007:**
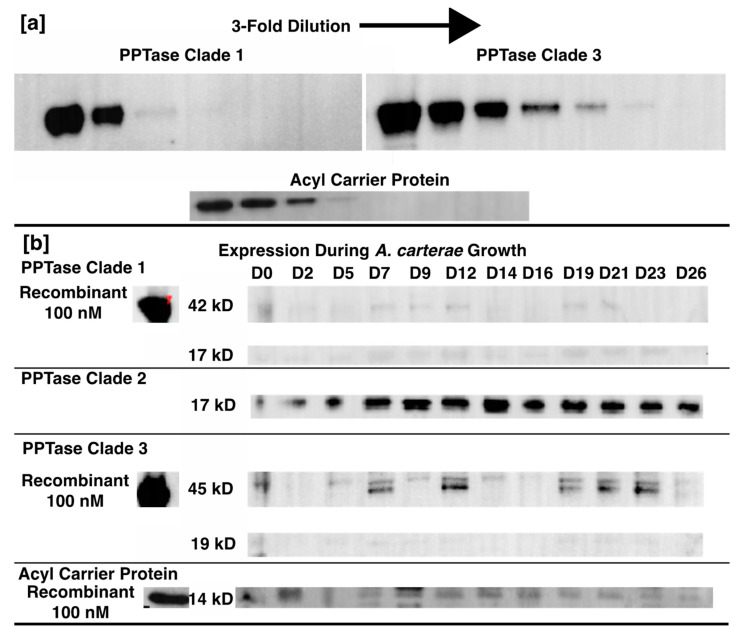
*Amphidinium carterae* protein quantities during growth. The upper pane (**a**) shows the 3-fold dilutions of the clade one and three phosphopantetheinyl transferases (PPTases) as well as the acyl carrier protein starting at 100 nM. The lower pane (**b**) shows the protein quantities of the clade one, two, and three PPTases as well as the acyl carrier protein. Each of the timepoints is labeled at the top of the pane with a “D” prefix for the day following CO2 control of pH. The left band is the 100 nM recombinant protein followed by the size of the band and finally the band images.

**Figure 8 marinedrugs-20-00581-f008:**
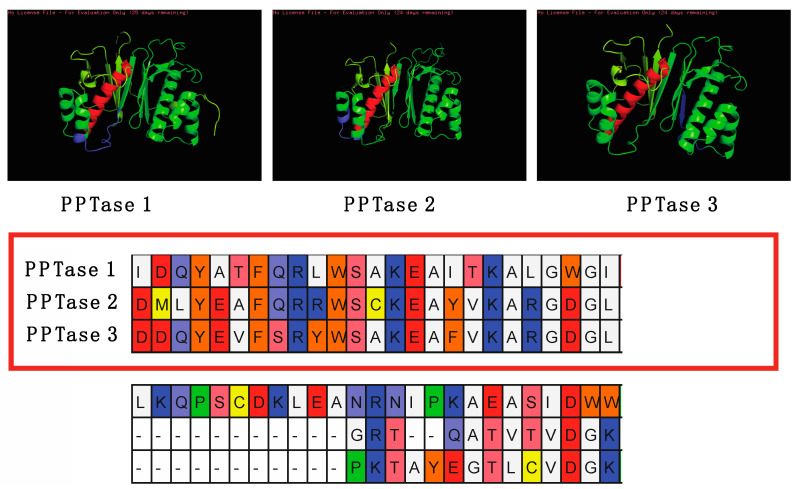
C-terminal alignment of the theoretical cleaved portion of *Amphidinium carterae* phosphopantetheinyl transferases. The three *A. carterae* PPTases are shown at the top with purple indicating the antibody epitope, red indicating the helix expected to be retained, and yellow indicating the beta sheet following the expected cleavage site based on the size of the lower band in western blots of *A. carterae* cultures. The alignment below starts with the conserved helix sequence marked with a red box followed by a short disordered region and then the beginning of the beta sheet.

**Figure 9 marinedrugs-20-00581-f009:**
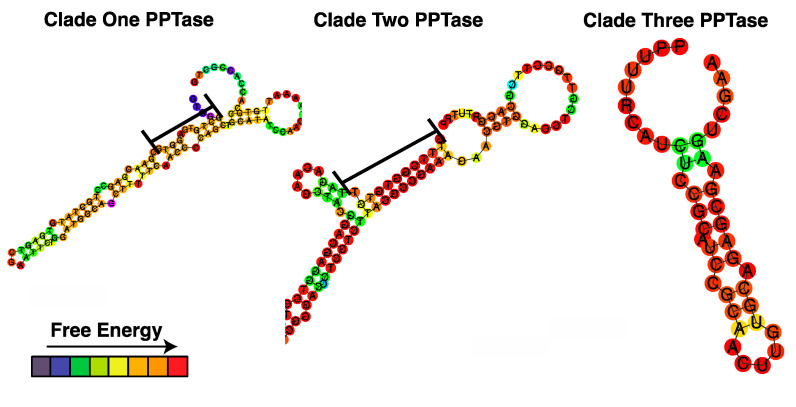
Folding structure of the *Amphidinium carterae* phosphopantetheinyl transferase 3′ untranslated regions. The 2-dimensionally rendered folding structure of the 3′ untranslated regions from the *A. carterae* phosphopantetheinyl transferases are shown. The colors indicate the relative free energy of each nucleotide from low in blue to high in red. Stem regions rich in guanidine and thymidine nucleotides are highlighted with a black bar.

**Figure 10 marinedrugs-20-00581-f010:**
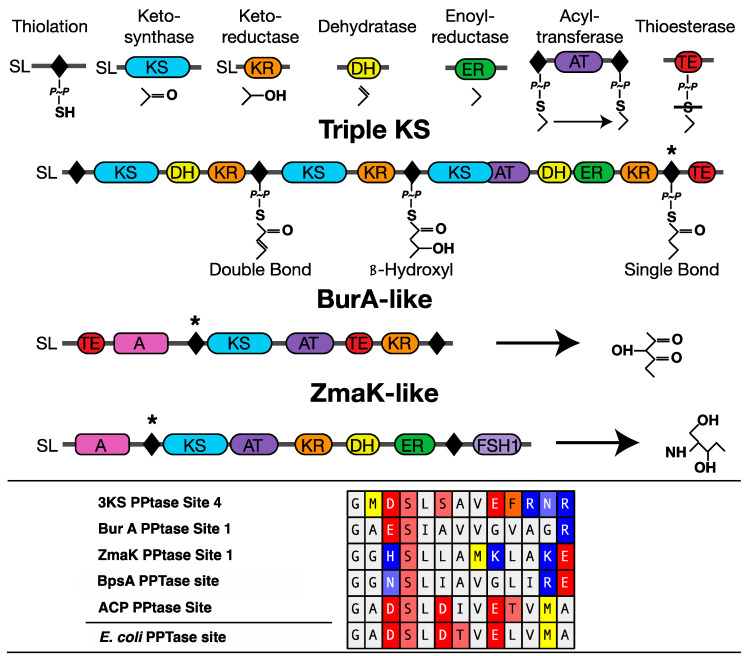
Domain arrangement of *Amphidinium carterae* transcripts containing thiolation domains used in this study. Individual modular synthase domains are shown at the top with example products for their reaction. In addition, Adenylation (A) and FSH1 serine hydrolases (FSH1) are shown for the multi-domain transcripts with examples of potential products included. The phosphopantetheinate group is shown as “P~P” with a single bond to a sulfur. “SL” refers to the dinoflagellate spliced leader sequence and is present if a spliced leader sequence has been verified. The bottom panel shown an amino acid alignments of the phosphopantetheinate binding domains for each of the transcripts with the thiolation domains marked with a * as well as thew acyl carrier protein. The *E. coli* acyl carrier protein sequence is additionally given as a reference.

**Figure 11 marinedrugs-20-00581-f011:**
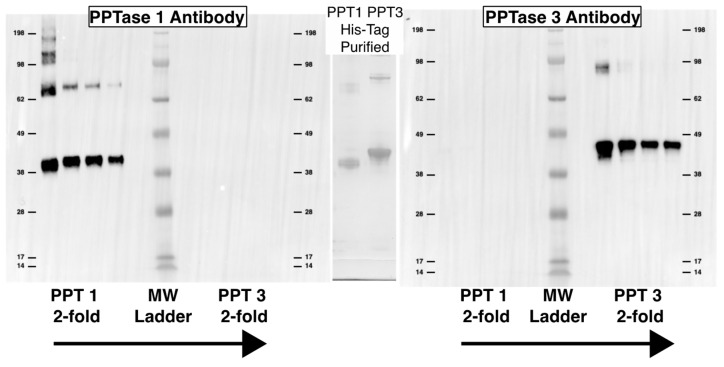
Purified protein controls for western blotting. Purified protein of *Amphidinium carterae* phosphopantetheinyl transferases (PPTase) used as western blotting standards are shown. The His-tagged purified proteins are shown in the middle as a Coomassie stained gel scaled to the same dimensions as the western blots. Each western blot has a 2-fold dilution of PPTase one on the left and PPTase three on the right with a molecular weight marker in the middle. The left blot was imaged with a PPTase one primary antibody while the right blot was imaged with a PPTase three primary antibody.

**Table 1 marinedrugs-20-00581-t001:** *Amphidinium carterae* proteins used to design antibodies.

Gene	Epitope	Molecular Weight (kD)	Isoelectric Point
PPTase Clade one	CAAPQLERGEGEDLS	39.5	5.24
PPTase Clade two	CVRQEGSLPARYEGA	39.5	7.98
PPTase Clade three	KGDRLHYKLSKGSGC	44.1	6.82
ACP	EEFEVDLPDEETTELKN	13.2	4.09

All sequences are from *Amphidinium carterae.*

**Table 2 marinedrugs-20-00581-t002:** Primers used in this study.

Primer Name	Sequence 5′ to 3′	Length	Annealing Temp. °C
PPT1_CDSF4 §	GCTTACAGTGGAGGCCCTACTTCCAATGGG	30	74.9
PPT2_CDSF4 §	TCCCTGCGGTGTCCAACTTCAAGCTTTACA	30	72.1
DTR2 §	CATCTTGCTAGCTCGCGATCTTGAAGTAGTC	31	72.1
Dino_SL §	TCCGTAGCCATTTTGGCTCTAA	21	59.5
anchoredDTR2 ☥	CATCTTGCTAGCTCGCGATCTTGAAGTAGTCTTTTTTTTTTTTTTTTTTTTS	52	41.9 *

Primers listed with an “§” were used in PCR amplification, while those with an “☥” were use in reverse transcription. The “anchored DTR2” annealing temperature denoted with a “*” is for the poly-T region only.

## Data Availability

Not applicable.

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
