# Peer review of "A Comparison of Dinoflagellate Thiolation Domain Binding Proteins Using In Vitro and Molecular Methods"

_marinedrugs, 2022, doi:10.3390/md20090581_

Round 1

Reviewer 1 Report

The current manuscript by Williams and colleagues compared thiolation domain binding proteins using various methods. Specifically the thiolation domains of dinoflagellate derived proteins were investigated via in-vintro and molecular mechanisms. The manuscript provides a well-structured investigation through expression, quantification, and sequence analysis towards understanding the role of these proteins. Figures are well crafted and support the general conclusions of the text. Only minor comments exist, specifically pertaining to figures and formatting.   

Figure 2: There is a blue box, could potentially be a highlight box, that is over the x-axis of the figure that appears to be out of place. This may also be an artifact of the review pdf.

Figure 8,9, etc.: Appears to be a different font than the rest of the text. Please standardize according to journal guidelines.  

Author Response

Reviewer 1:

1: Figure 2; The blue box appears to be a conversion problem from the PDF. The image has been changed to a TIFF format to remove this issue.

2: Figures 8 and 9; Fonts for the alignment in figure 8 have been changed to the standard format but the nucleotides in figure 9 are part of an exported image from a website and unfortunately cannot be changed. Some other fonts have been altered to be more consistent in size and format.

Reviewer 2 Report

This is an interesting manuscript on the in-vitro heterologous expression of dinoflagellate proteins from Amphidinium carterae involved in the synthesis of indigoidine. All the experiments have been carried out properly, and the results are clearly presented. I have just a few comments listed below.

In general, the author should indicate more frequently that experiments have been carried out using Amphidinium carterae as model species, instead of “dinoflagellate”.

I wonder if the author have tried to blast the transcripts containing thiolation domains against the A. carterae transcriptome obtained in nitrogen-starved and -replete culturing conditions by Lauritano et al. (2017). The authors provided a list of differentially expressed genes with KEGG pathways and also reported the occurrence of several KS genes.

Lauritano, C., De Luca, D., Ferrarini, A., Avanzato, C., Minio, A., Esposito, F., & Ianora, A. (2017). De novo transcriptome of the cosmopolitan dinoflagellate Amphidinium carterae to identify enzymes with biotechnological potential. Scientific reports, 7(1), 1-12.

Another point (mostly a curiosity) is about the phylogenetic relationships among these three PPT transcripts. The authors often refer to clades, but it is not clear on the basis of what. Furthermore, since these transcripts are all functional, a few words more on their possible evolutionary significance could improve the discussion. For instance, do the author think that they constitute a redundancy mechanism?

Line 26 = I wonder why PKS has been inserted among keywords and it is never cited in the text, except as KS at line 103

Line 32 = please change as “like algae” or “like other protists” otherwise it seems that dinoflagellates are algae

Line 61 – 62 = do the author mean that few researchers work on such domains because they are involved in the last step(s) of biosynthetic pathways? If not, please clarify it.

Line 168 = why the authors refer to a clade? Is there a phylogeny of these transcripts? Please explain

Line 325 = some letters of the word “frames” are underlined

Line 340 = please use the MDPI style for in-text citations

Line 454 = there is no table title. Please remove bold for primer name PPT1

Figure legends are not in the correct style format

Author should report full citation (brand, city, country) for instruments/materials

Author Response

Reviewer 2:

1: Use of Amphidinium carterae; A. carterae has been specified whenever specific results have been referenced especially in the abstract and introduction as well as the discussion.

2: Gene expression in A. carterae during nitrogen starvation; Unfortunately, the thiolation domains are so similar that they are very difficult to differentiate apart from being able to identify the ACP-like versions. Thus, transcriptome to transcriptome comparisons, especially for different strains, don’t work so well for these genes. Although the copy number is high, the ketosynthases are better candidates for differential expression since they are more diverse and was the original reason we read the referenced publication. This utility of using ketosynthases based on the presented findings in the 2017 paper have been added to the discussion and the references.

3: The three clades of PPTases; This is a tricky one since there doesn’t seem to be any consistency about whether or not a particular PPTase is retained in a dinoflagellate lineage with the exception of clade 3 and their origin is unclear since they are clearly not in any sequenced syndinian dinoflagellate. This uncertainty and the possibility of a redundancy mechanism have been added to the discussion section on the PPTases, specifically with respect to ensuring that the ACP is fully phosphopantetheinated.

4: The “PKS” nomenclature; Good point! Although this response may be disappointingly political. While we think this work may be of interest to researchers who study ketosynthase-based natural product synthesis, the term “PKS” itself isn’t very informative as it tries to differentiate from NRPS that is often part of the same pathways including the BurA and ZmaK-like transcripts in this study that have both adenylation and ketosynthase domains. Thus, we tried to focus on the domains themselves rather than bin them into PKS or NRPS terminology. “NRPS” was not included in the keywords because it is a less common term and we did not specifically investigate adenylation or condensation domains. We are certainly amenable to removing “PKS” as a keyword if the reviewer feels it is inappropriate.

5: Dinoflagellates as algae; We have changed this to “like other protists” as well as removed “photosynthetic” since species like Crypthecodinium as fit this description.

6: Thioesterase genes as a candidate for research; This is more an argument about the utility of this gene in general as a justification for not investigating it in this study rather than a statement about the current body of research. The sentence has been modified as follows to better state this “The thioesterase gene was not chosen for this study because it acts to terminate a series of processes and given the size of the A. carterae toxin a traceback of intermediate products was not feasible, although knockouts in other systems have proven fruitful [36; 37]”

7: PPTase clade; The reference giving the phylogenetic context for the clades was not properly described in the introduction and has been amended to specifically refer to the establishment of the three clade nomenclature. Additionally, the first reference to a “clade” in each section includes this reference.

8: underlined letters; Thank you for catching this.

9: in-text citations; citation formats have been corrected and placeholders removed.

10: Table titles; Titles have been added to all tables and bold fonts have been corrected for consistency.

11: Figure legend formatting; Figure legends have been corrected for the use of bold fonts and case.

12: Instrument/Material citations: Citations have been corrected to have brand, city, and country on first mention.